

# A review of behavioral testing in decapod shrimp (Caridea) and prawns (Dendrobranchiata) with applications for welfare assessment in aquaculture

Dana L. M. Campbell and Caroline Lee

Agriculture and Food, Commonwealth Scientific and Industrial Research Organisation (CSIRO), Armidale, New South Wales, Australia

## ABSTRACT

Evolving societal expectations are driving increasing interest in the welfare of decapod crustaceans, such as prawns and shrimp, grown in aquaculture. A key aspect of understanding an animal's welfare-related needs is through assessing their behavior to determine how the animal is perceiving and interacting with their environment. Behavioral testing has been applied to livestock animals for decades, providing insight into their wants and needs to guide housing structure design and husbandry practices that improve their welfare. This review collated studies that have applied behavioral testing, primarily at the individual level, to decapod shrimp and prawns in the Dendrobranchiata and Caridea sub- and infra-orders respectively. This review aims to understand the types of assessments that can be successfully applied to these taxa, and what the results of testing may be able to inform us about in regard to the welfare of these species. While the sentience capabilities of these decapod taxa is still under debate, the behavioral testing applied to date across varying species demonstrates they exhibit preferences across multiple contexts, individual differences indicative of personality, cognitive capabilities, and behavioral indicators consistent with negative affective states. There is scope to learn from livestock welfare assessment using behavioral testing and increase the research focused on penaeid shrimp and prawn species of aquaculture interest. Application and validation of new behavioral tests can guide system optimization for aquaculture shrimp and prawns in relation to the welfare of the animals.

## INTRODUCTION

As the global drive to improve the welfare of animals that we farm continues, societal expectations are evolving. Welfare concerns have expanded from terrestrial livestock animals and farmed fish to include the welfare of decapod crustaceans, such as prawns and shrimp grown in aquaculture (*Albalat et al., 2022*; *Pedrazzani et al., 2023*, *2024*; *Wuertz, Bierbach & Bögner, 2023*). Considering the numbers of individual animals grown in aquaculture settings, there are an estimated 440 billion animals raised each year (*Romero Waldhorn & Autric, 2022*), and they are being produced at an upward trajectory

Corresponding author
Dana L. M. Campbell,
Dana.Campbell@csiro.au

(*Villarreal, 2023*). Based on numbers alone, the scope for potential welfare, and associated production impacts, is substantial. However, the extent of welfare related scientific evidence to guide housing and management practices is comparatively less than what has been produced to date for domesticated terrestrial species (*Albalat et al., 2022*). This discrepancy is in part due to the younger age of shrimp and prawn industries relative to livestock farming; in part due to the challenges around assessment of vast numbers of smaller-sized animals that live underwater; and in part due to global debate around the sentience of these species. It is currently undetermined as to whether these animals have the capacity to experience negative affective states such as pain, fear, and distress that could compromise their welfare state (*Birch et al., 2021*).

*Birch et al. (2021)* reviewed the scientific evidence for decapod sentience across eight specified sentience criteria to conclude that any current evidence of sentience to date comes from the larger species such as lobsters, crayfish, and crabs. In comparison, smaller species, such as prawns and shrimp, have had very little research conducted, resulting in an 'absence of evidence' rather than 'evidence of absence' of sentience capabilities (*Birch et al., 2021*). Despite the sentience dispute (*Diggles, 2019*; *Diggles et al., 2024*), in 2022 the UK government extended the scope of their Animal Welfare (Sentience) Act to include decapods as sentient beings. Good welfare practices in commercial production settings can be guided by ethical principles erring on the side of caution around sentience capacity (*Birch, 2017*), and will also lead to improved animal health and performance regardless of any emotional perceptions of the animal (*Wuertz, Bierbach & Bögner, 2023*).

A key aspect of understanding an animal's welfare-related needs, is through assessing their behavior (*Dawkins, 2004*). Behavioral observations can document how the animal is perceiving and interacting with their environment. This then determines if they are healthy and if the provided resources are meeting their needs (*Dawkins, 2004*). Prawn and shrimp behaviors can be observed to guide commercial housing and management practices and/or be quantified as a metric to assess the animals' welfare status, as detailed in recent reviews focused on commercially relevant penaeids (*Albalat et al., 2022*; *Bardera et al., 2019a*; *Pedrazzani et al., 2023*). However, there is great scope for more research on the behavior of prawns and shrimp in relation to welfare improvements in aquaculture. In particular, we have identified behavioral testing at the individual level as an area of future research focus for understanding prawn and shrimp welfare needs. Behavioral testing has been applied for decades across countless contexts in livestock welfare research, to inform on factors such as preferences for resource types, temperament, social drivers, discriminatory capabilities, cognition, and affective state (*e.g.*, *Baciadonna & McElligott, 2015*; *Duncan, 2005*; *Forkman et al., 2007*). This knowledge then guides the way the animals are housed and managed. There are also new behavioral tests continually being validated to objectively measure how an animal is perceiving their environment, and factors that influence that perceptive experience (*e.g.*, attention bias: *Monk, Campbell & Lee, 2023*, startle tests: *Salvin et al., 2020*). Applying individual behavioral testing and validating new behavioral tests can expand the scientific knowledge on decapod species to be drawn on in the context of welfare measurement and improvement in aquaculture.

The decapod order of animals contains, by recent estimates, over 17,000 species and includes all 10-legged crustaceans (*De Grave et al., 2023*; *Wolfe et al., 2019*). If focusing on only the prawns and shrimp, there are two suborders (Dendrobranchiata, Pleocyamata) and three infraorders (Stenopodidea, Caridea, (Procaridea—primarily fossil species)) that include thousands of anatomically diverse species spanning across freshwater and marine habitats globally (*De Grave et al., 2023*; *Wolfe et al., 2019*). The terms 'prawn' and 'shrimp' are often used interchangeably depending on regional nomenclature preferences, but under the decapod order, prawns are the Dendrobranchiata species, and shrimp are the Caridea and Stenopodidea species. In this manuscript we utilize the same term as applied in each reviewed study with the animal suborder/infraorder stated per species. Despite the thousands of prawn and shrimp species that exist, there are only two that comprise the majority of commercial production globally, the Pacific whiteleg shrimp (*Litopenaeus vannamei*) and the Giant tiger prawn (*Penaeus monodon*) of the Penaeidae family (Dendrobranchiata) (*Cai et al., 2017*; *FAO, 2024*). In their natural habitats, these animals are tropical marine invertebrates as adults, with the larvae and juveniles found in brackish waters (*Dugassa & Gaetan, 2018*; *Motoh, 1985*). These species are typically bottom dwellers that mate and then the female spawns into the water (*Motoh, 1985*; *Primavera, 1979*; *Yano et al., 1988*). In commercial production, male and female broodstock can be wild caught or domesticated (*Marsden et al., 2013*; *Ren et al., 2020*), with the post-larvae raised in grow-out ponds at varying degrees of farming intensity (*El-Sayed, 2020*; *Emerenciano et al., 2022*). As with any animal species for commercial production, understanding the animal's behavioral ecology and needs can guide optimization of production systems, improving both production efficiency and the welfare of the animal.

The aim of this review is to collate the studies to date that have applied behavioral testing to decapod shrimp and prawns across varying contexts relevant to welfare assessment. The focus is primarily individual behavioral assessments, to understand the types of tests that can be successfully applied to these taxa in research contexts, and what the testing may be able to inform on in regard to individual animal welfare. The information obtained from the tests across studies is typically species specific. However, demonstrating that the animals can perform in these test situations would allow similar tests to be applied to relevant commercial species by researchers in aquatic animal welfare science. We compare the individual behavioral assessments of decapod prawns and shrimp to similar tests that have been commonly applied to livestock species. This comparison highlights the potential of these tests to inform on ways to optimise housing and management of commercial species, and methods for assessing and improving welfare state in aquaculture. Researchers working in aquaculture welfare can utilize the information in this manuscript to increase the application of these tests which can then provide management guidance to shrimp and prawn industries for system optimization.

## SURVEY METHODOLOGY

'Google Scholar' was used to conduct the literature searching with a wide range of search terms. This wide scope was set to identify different test contexts, as the aim was to understand the types of tests that have been successfully applied to these animals, drawing
on the authors' knowledge of the typical individual livestock behavioral testing under a welfare context. New terms were added to the search as further testing contexts were identified through citations. Search terms included: 'shrimp OR prawn fear'; 'shrimp OR prawn anxiety'; 'shrimp OR prawn associative learning'; 'shrimp OR prawn cognition learning'; 'behavior OR behaviour demand prawns OR shrimp motivation'; 'behavior OR behaviour prawn OR shrimp motivation'; 'prawn OR shrimp personality'; 'shrimp OR prawn behavior OR behaviour preference'; 'colour OR color preference prawn OR shrimp'; 'prawn OR shrimp feed preference behavior OR behaviour'; 'prawn OR shrimp feed Y-maze'; 'shrimp OR prawn mating test'; 'shrimp OR prawn diet behavior OR behaviour preference'; 'shrimp OR prawn feed preference behavior OR behaviour'; 'shrimp OR prawn feed choice test'; 'shrimp OR prawn thermal preference'; 'shrimp OR prawn salinity preference'; 'shrimp OR prawn light preference'; 'shrimp OR prawn substrate preference'; 'shrimp OR prawn shelter preference'. The downloaded articles were sorted through to include only those conducted on decapod shrimp or prawns, and testing situations focused primarily at the individual level, or small groups in a laboratory setting where there was a behavioral test conducted in a set time period. Only articles with behavioral measures were included. Other articles were obtained through the reference lists of those that were downloaded, until a saturation point was reached where the same articles were being identified in any follow-up searches.

## Preference/choice testing, discrimination, and aversion

Preference or choice testing is a widely used paradigm for assessing if an animal can distinguish between two (or more) different objects, environments, or conspecifics and if they may have a preference for, or an aversion to, one resource/context *vs.* the other. This type of testing has been applied for decades to understand what resources/contexts an animal may desire or avoid, to aid in optimising housing design, management practices, and welfare assessment for livestock animals (*Bateson, 2004*; *Dawkins, 2004*). This same testing paradigm is the most commonly applied behavioral test to decapod shrimp and prawns across varying contexts, which are grouped here into environmental, food, and social preferences. The species assessed and behavioral tests applied are summarized in Table 1.

## Environmental preference

Sensory discrimination and choice can be used to understand both the capabilities of the animal, as well as what type of environmental conditions they may prefer. For shrimp and prawns, environmental preference testing has been conducted for various factors including shelter provision, substrate type, colors, water temperature, and salinity. *Lammers, Warburton & Cribb (2009a)* examined environmental preferences in *Macrobrachium australiense* (Caridea) across a series of trials comparing shelter vegetation and open areas. The prawns were assessed for their choice comparing an open graveled area *vs.* varying densities of artificial refuge stalks; the lowest *vs.* the highest density vegetation simultaneously; and two different vegetation densities separated by an open area in the middle. These tests showed that the prawns preferred to take refuge in the vegetation,

**Table 1 A summary of the different types of behavioral tests that have been applied across varying prawn and shrimp species detailing what they measure and their welfare application.**

| Behavioral test | What the test measures | Species applied to | Citations | Welfare application |
|---|---|---|---|---|
| Two or more choice preference/ avoidance/discrimination tests including environment choice, resource choice, sensory choice, toxicity discrimination. | Can the animal distinguish between and display a choice/ preference for one presented option over another. Preference may be positive choice, or avoidance of a negative stimulus. | *Macrobrachium rosenbergii* | *Stephenson & Knight (1982), Kawamura et al. (2016, 2017, 2020), da Costa et al. (2023)* | Used to determine if animals differ in strength of desire for one option over another. A method of asking animals 'what they want', or 'what they do not want'. |
| | | *Macrobrachium australiense* | *Lammers, Warburton & Cribb (2009a)* | |
| | | *Penaeus merguiensis* | *Meager et al. (2005)* | |
| | | *Palaemon elegans* | *Chapman, Hegg & Ljungberg (2013)* | |
| | | *Macrobrachium nobilii* | *Mariappan & Balasundaram (2003)* | |
| | | *Neocaridina davidi* | *Plichta et al. (2021)* | |
| | | *Crangon crangon* | *Evans, Lyes & Lockwood (1977), Reiser et al. (2013), Reiser, Herrmann & Temming (2014), Reiser et al. (2017)* | |
| | | *Palaemon varians, Litopenaeus vannamei* | *Redondo-López et al. (2023)* | |
| | | *Litopenaeus vannamei* | *Hernández et al. (2006), González et al. (2010)* | |
| | | *Penaeus monodon* | *Chen & Chen (1999)* | |
| | | *Farfantepenaeus aztecus, Litopenaeus setiferus* | *Doerr, Liu & Minello (2016)* | |
| Feed preference including palatability and attraction. Predominantly Y-maze choice testing. | What food types are more palatable, what can attract an animal to feed, how food type affects feeding behavior. Tests primarily applied from a nutritional/production perspective to date. Potential for application to understand welfare-related factors such as dietary choice or cognitive stimulation through feed. | *Litopenaeus vannamei* | *da Silva et al. (2013), Nunes et al. (2006), Ramírez et al. (2017), Derby et al. (2016)* | Used to determine preferred food types, palatability and nutritional content. Potential for understanding impacts of dietary choice or cognitive stimulation through feed to ensure the animals' nutritional needs are met as well as affective needs through food. |
| | | *Litopenaeus stylirostris* | *Suresh, Kumaraguru vasagam & Nates (2011)* | |
| | | *Macrobrachium tenellum* | *Montoya-Martínez et al. (2018a, 2018b)* | |
| | | *Macrobrachium rosenbergii* | *Das et al. (2019)* | |
| | | *Penaeus monodon* | *Hartati & Briggs (1993)* | |
| | | *Palaemonetes antennarius* | *Constantini & Rossi (2001)* | |

(Continued)

| Behavioral test | What the test measures | Species applied to | Citations | Welfare application |
|---|---|---|---|---|
| Social discrimination and preferences: includes shoaling preferences, predator recognition, individual recognition, familiar/unfamiliar conspecific recognition. | Do the animals show individual recognition, conspecific/heterospecific recognition/avoidance, mate preferences, and social contact preferences. | *Palaemon elegans* | *Chapman, Hegg & Ljungberg (2013)* | Used to understand the social discriminatory capabilities of a species/individual and determine parameters that define positive and negative social interactions to be able to address the social needs of an animal. |
| | | *Palaemon elegans, Crangon crangon* | *Evans, Finnie & Manica (2007)* | |
| | | *Paratya australiensis* | *Bool et al. (2011)* | |
| | | *Caridina typus, Palaemon affinis* | *Brooker & Dixson (2017)* | |
| | | *Rhynchocinetes typus* | *Díaz & Thiel (2003), Thiel & Hinojosa (2003)* | |
| | | *Alpheus heterochelis* | *Rahman, Dunham & Govind (2002), Rahman, Dunham & Govind (2001), Mathews (2003), Ward et al. (2004)* | |
| | | *Lysmata debelius* | *Rufino & Jones (2001),* | |
| | | *Macrobrachium rosenbergii* | *Barki, Karplus & Goren (1990, 1991)* | |
| Resource prevention: blocking substrate access with a transparent barrier. | Behavioral and/or physiological responses to having access to a specific resource prevented. | *Crangon crangon* | *Siegenthaler et al. (2018)* | Used to assess the value of specific resources to the animal to be able to provide resources they want/need. |
| Resource competition: includes dominance and aggression around shelter resources, food, and mates. | Agonistic interactions between individuals over specific resources and how competition and access can be influenced by dominance hierarchies. | *Macrobrachium australiense* | *Lammers, Warburton & Cribb (2009b)* | Used to understand how social factors affect resource access to be able to provide adequate resources to all individuals and minimise stress/injury resulting from resource competition. |
| | | *Macrobrachium rosenbergii* | *Barki, Karplus & Goren (1992)* | |
| | | *Macrobrachium nobilii* | *Mariappan & Balasundaram (2003)* | |
| | | *Litopenaeus vannamei* | *Bardera et al. (2021)* | |
| | | *Rhynchocinetes typus* | *Dennenmoser & Thiel (2006), Correa et al. (2003)* | |
| | | *Alpheus heterochaelis* | *Rahman, Dunham & Govind (2004)* | |

| Table 1 (continued) | | | | |
|---|---|---|---|---|
| Behavioral test | What the test measures | Species applied to | Citations | Welfare application |
| Cognition and learning: includes Y-maze learning, spatial and manipulation tasks utilising transparent barriers. | Ability to complete and speed of learning specific tasks that require cognitive capabilities such as spatial awareness, discrimination, and memory. | *Macrobrachium acanthurus*<br><br>*Palaemon elegans*<br><br>*Palaemon serratus* | *Ventura & Mattei (1977)*<br><br>*Duffield, Wilson & Thornton (2015)* | Used to understand cognitive capabilities to determine how an animal may adapt to their housing environment, whether they possess the capabilities to learn tasks that may be asked of them, how resource provision (or lack thereof) can affect cognitive development and what cognitive stimulation is needed to avoid negative states and/or development of abnormal behaviors. |
| Affective state, temperament, and personality. Includes: open field test, shelter-seeking, escape-responses, fright (startle) test, food *vs.* shelter trial. | | *Penaeus monodon*<br><br>*Neocaridina denticulata* ssp., *Palaemon pacificus*<br><br>*Neocaridina denticulata*<br><br>*Limnocaridina latipes*<br><br>*Crangon crangon*<br><br>*Palaemon elegans* | *Harayashiki et al. (2016)*<br><br>*Takahashi (2022)*<br><br>*Rickward, Santostefano & Wilson (2024)*<br><br>*N. denticulata* only; *Takeuchi, Shoko & Michio (2008)* (both species)<br><br>*Arnott, Neil & Ansell (1999)*<br><br>*Chapman, Hegg & Ljungberg (2013), Maskrey et al. (2018)* | Used to understand the animal's perception of their environment and how varying contexts can affect their emotional state as well as how individual differences in personality can impact resource use, social interactions, and other experiences. This can inform on husbandry to minimise negative states and increase positive states. |

regardless of the density. When the prawns were making a choice between two different refuge patches with a 'cost' of crossing an open area to reach the other refuge, the time taken to leave the first refuge patch was shorter if the alternate patch was of higher density than what the prawns had been initially placed in. The latency to leave the first refuge patch, and the number of crossings made of the open area was also affected by prawn size, with larger individuals showing behaviors equating with higher vulnerability—*i.e.*, quicker to leave if the starting patch was low quality, and fewer crossings made (*Lammers, Warburton & Cribb, 2009a*). *Meager et al. (2005)* also assessed habitat preferences of groups of 10 juvenile *Penaeus merguiensis* (Dendrobranchiata) prawns in a circular tank with four areas of varying degrees of shelter. As predicted, prawns were observed more often in the habitats that provided the most refuge shelter, although choices were not affected by prawn size (*Meager et al., 2005*). *Chapman, Hegg & Ljungberg (2013)* observed the choice that individual *Palaemon elegans* (Caridea) made between a vegetated side of a

test aquarium and an open sand side. They quantified how many times the individual changed between the two habitat types as a measure of 'exploration', to be correlated with other behavioral test measures (see section on '*Affective state, temperament, and personality*'). Preference for PVC pipe shelters, including their color, has been observed in *Macrobrachium nobilii* (Caridea; *Mariappan & Balasundaram, 2003*). The shelter choice of these prawns was dependent on the size of the individual, with shelter preferred over an open area, and darker colored shelters occupied more often (*Mariappan & Balasundaram, 2003*). Preferences for different substrate colors, as well as fine or coarse checkerboard patterns enclosing experimental test dishes have also been assessed in different color morphs of *Neocaridina davidi* (*Plichta et al., 2021*).

Beyond shorter term (less than 24 h, typically minutes to a couple of hours) behavioral testing of individuals or small groups of animals for habitat or resource selection, choices can also be quantified over longer periods. Testing across several days or weeks allows observations of where animals may prefer to spend their time across an extended duration, as well as how resource preferences may affect other exhibited behaviors (*e.g.*, *Balasundaram, Jeyachitra & Balamurugan, 2004*; *Carvalho-Batista et al., 2023*; *Luchiari, Marques & Freire, 2012*; *Ouellette et al., 2003*; *Santos et al., 2015*; *Smith & Sandifer, 1979*). Short-term testing often requires an acclimation period to the test arena prior to observations commencing. The decision to test across several hours or several days will depend on the resource being assessed, the expected use pattern (*e.g.*, will animals use the resource immediately, or only at certain times of day) and the research question being asked (*e.g.*, changes in preferences across time).

There have also been a series of studies on *Macrobrachium rosenbergii* (Caridea) to observe colored bead preferences in larvae (*Kawamura et al., 2016*), shelter color preferences in post-larvae (*Kawamura et al., 2017*), and how background preferences change between larvae and post-larvae life stages (*Kawamura et al., 2020*). However, these tests were not conducted at the individual level but instead predominantly in groups of 100+ individuals which may be a more feasible experimental strategy for these smaller sized life stages. Testing of individual *M. rosenbergii* also showed shelter color preferences changed from juveniles to adults (*da Costa et al., 2023*). These studies across life stages can indicate the importance of environmental considerations aligned with changes in physiology and development (*da Costa et al., 2023*; *Kawamura et al., 2020*).

The choice testing paradigm can be applied to test the perception of water quality related environmental parameters, including avoidance of toxic environments. The effects of adverse chemicals were examined in *Crangon crangon* (Caridea) where *Evans, Lyes & Lockwood (1977)* assessed how oil dispersants reduced the ability of the shrimp to be able to detect the Y-maze arm that contained a feed attractant. *Redondo-López et al. (2023)* looked at avoidance of different concentrations of copper toxicity by *L. vannamei* and *Palaemon varians* (Caridea) shrimp. They demonstrated that both species avoided the chamber compartments with the higher copper concentrations, spending their time in the lower concentrated, less aversive compartments (*Redondo-López et al., 2023*). Gradient choice chambers have also been used to assess thermal and/or salinity preferences by

observing which chamber the animals choose to occupy (*Crangon crangon*: *Reiser et al., 2013*; *Reiser, Herrmann & Temming, 2014*; *Reiser et al., 2017*; *Litopenaeus vannamei*: *González et al., 2010*; *Hernández et al., 2006*; *Penaeus monodon*: *Chen & Chen, 1999*; *Macrobrachium rosenbergii*: *Stephenson & Knight, 1982*; *Farfantepenaeus aztecus* and *Litopenaeus setiferus*: *Doerr, Liu & Minello, 2016*). Measuring the choice for combinations of environmental parameters can provide guidance for optimizing commercial conditions, particularly in smaller holdings such as broodstock facilities, where there is greater scope for environmental control.

## Preventing access to resources

Another method of assessing the value of a particular resource to an individual is through preventing resource access and then quantifying the physiological and/or behavioral impacts on the animal. This type of behavioral assessment has been applied across many different livestock species, such as preventing laying hens from accessing perches for roosting at night (*Olsson & Keeling, 2000*), or preventing dairy cows from feeding and lying (*Cooper, Arney & Phillips, 2008*). To the authors' knowledge, only one study has assessed resource prevention in *C. crangon* shrimp (*Siegenthaler et al., 2018*). Individuals were assessed to determine how camouflage coloring was impacted by different treatment conditions of light, sediment color, and burying access. Naturally, these epibenthic animals will bury themselves in sand up to their eyes and antennae (*Pinn & Ansell, 1993*) indicating the importance of the sediment resource. When sediment burying ability was prevented *via* a transparent plastic barrier, the shrimp turned to a darker color, irrespective of the background color they were transferred to (black or white). This was interpreted as a likely 'stress response' indicating physiological changes, that could be used to quantify the value of the resource (*Siegenthaler et al., 2018*). To build upon this, there is scope to validate the application of behavioral demand tests in these animals as another strategy for assessing resource value. Behavioral demand measures the motivation of an individual to access a resource, and has been validated across many livestock species (*Jensen, Pedersen & Ladewig, 2004*). Behavioral demand testing quantifies the effort invested into reaching a resource, such as squeezing through small spaces (*Cooper & Appleby, 1996*) or pushing weighted doors (*Cooper & Mason, 2000*). These types of assessments have shown, for example, that food is just as valued as brush access in dairy cows (*McConnachie et al., 2018*).

## Feed preferences

The choice paradigm has also been applied in several studies to measure food preferences of various species (Dendrobranchiata: *P. monodon, L. vannamei, Litopenaeus stylirostris*, Caridea: *M. rosenbergii, Macrobrachium tenellum, Palaemonetes antennarius*). These studies have primarily been conducted from a nutritional and production viewpoint, testing the choices made for feed related factors such as palatability and attraction (*da Silva et al., 2013*; *Das et al., 2019*; *Montoya-Martínez et al., 2018b*; *Nunes et al., 2006*; *Ramírez et al., 2017*). The predominant preference paradigm is a Y-maze two-choice design

(see *Montoya-Martínez et al., 2018a* for an evaluation of different maze designs in *M. tenellum*). However, some studies have used choice aquaria divided into different sections (*Hartati & Briggs, 1993*; *Montoya-Martínez et al., 2018a*), presented more than two choices simultaneously (*Constantini & Rossi, 2001*; *Suresh, Kumaraguru vasagam & Nates, 2011*), or looked at interactions with an airstone delivering chemical feed attractants (*Derby et al., 2016*). Depending on the set-up of choice, animals are tested in small groups (*e.g.*, up to 25 individuals), or tested individually. Behaviors measured include variables such as time to approach the feed and consumption or rejection of the feed offered. The results of these studies collectively demonstrate that the animals often do show preferences or rejections, and will make choices in this context (*Constantini & Rossi, 2001*; *da Silva et al., 2013*; *Das et al., 2019*; *Montoya-Martínez et al., 2018b*; *Nunes et al., 2006*; *Ramírez et al., 2017*; *Suresh, Kumaraguru vasagam & Nates, 2011*). There is scope to expand these types of studies to assess preferences for food-related factors that may have welfare components, such as cognitive stimulation through feed and/or dietary choice. This type of individual testing could also aid in understanding other factors that will influence feeding behaviors. Variables such as feed deprivation and moult status (*Bardera et al., 2019b*), sex (*Bardera et al., 2020*), or social competition (see section on '*Resource competition*') can all have implications in ensuring optimal management at different phases across commercial production cycles.

## Social discrimination and preferences

The choice paradigm has also been used to measure social preferences or shoaling tendencies. Social choice testing includes many studies conducted on Caridean species from a behavioral ecology perspective aimed toward understanding social recognition/discrimination (*Chak, Bauer & Thiel, 2015*). *Chapman, Hegg & Ljungberg (2013)* assessed *P. elegans* and *Evans, Finnie & Manica (2007)* assessed both *P. elegans* and *C. crangon* measuring the time spent in proximity to stimulus conspecifics in a test chamber. There were differences in the strength of shoaling preference across the two species tested by *Evans, Finnie & Manica (2007)*. *C. crangon* showed a weaker affiliation to the larger group of five conspecifics, *vs.* time spent in proximity to only a single conspecific stimulus prawn relative to the clear group preference of *P. elegans* in the same test set-up. *C. crangon* individuals also preferred to shoal in proximity to larger-sized stimulus individuals, regardless of whether their own body size was small or large (*Evans, Finnie & Manica, 2007*). The results of shoaling tests in *Chapman, Hegg & Ljungberg (2013)* were correlated with other behavioral tests to understand the presence of behavioral syndromes looking at consistency between tests and across time (see sections on '*Environmental preferences*', and '*Affective state, temperament, and personality*' for other tests applied by *Chapman, Hegg & Ljungberg (2013)*.

*Bool et al. (2011)* used a Y-maze choice test set-up to investigate whether the glass shrimp *Paratya australiensis* (Caridea) discriminated against and avoided water infused with a predatory fish scent over control aged tap water. They assessed the shrimp both when they were naïve and when they were exposed to the predatory fish for 3 days prior to the testing. While shrimp chose to enter one of the Y-maze arms at random, those shrimp

that had been exposed to the predator significantly increased the time taken to choose an arm compared with shrimp that had not been exposed (*Bool et al., 2011*). The Y-maze test in this context measured avoidance of, rather than preference for, an arm (see *Feed preferences* section). *Brooker & Dixson (2017)* also used a two-chambered choice set-up to look at the scent discriminatory capabilities of two shrimp species (*Caridina typus* and *Palaemon affinis*; Caridea). They determined if they showed preferences for conspecific cues, or avoidance of predatory heterospecific cues or non-predatory heterospecific cues relative to unscented control water. By assessing relative time spent in the water flume with each scent, a significant preference was demonstrated in both species for conspecific cues, and significant avoidance of predatory heterospecific cues (*Brooker & Dixson, 2017*).

Results from social recognition/discrimination laboratory tests has improved understanding of the behavioral ecology of several shrimp species. For example, the tests have shown that females of *Rhynchocinetes typus* (Caridea) show preferences for dominant males (*Díaz & Thiel, 2003*) and will exhibit cryptic choice against subordinate males by accepting their sperm, but then delaying spawning and removing the spermatophores (*Thiel & Hinojosa, 2003*). *Alpheus heterochaelis* (Caridea) shrimp will exhibit stable size assortative pairing (*Rahman, Dunham & Govind, 2002*) and recognize a former mate against an unfamiliar shrimp (*Rahman, Dunham & Govind, 2001*). In Y-maze testing, males have been shown to differentiate between chemical cues in the water from males and females varying in their molt status (*Mathews, 2003*). They have also shown recognition and preference for familiar same-sex conspecifics over unfamiliar individuals in a Y-maze test paradigm, which was likely due to differentiating between different chemical signatures (*Ward et al., 2004*). Finally, individual mate recognition relative to a stranger has been documented through laboratory behavioral preference testing in *Lysmata debelius* (Caridea) shrimp (*Rufino & Jones, 2001*). These tests collectively highlight how these animals have evolved to maximize their fitness through mate selection and reproductive processes. Understanding of the natural biology of mating systems through behavioral testing in the laboratory could be applied to improve breeding within captive prawn and shrimp aquaculture, using either wild-caught or domesticated broodstock. Other social relationships, such as dominance, can be obtained from observations of aggressive interactions between individuals within small groups in a test arena (*M. rosenbergii*: *Barki, Karplus & Goren, 1990*, *1991*). These hierarchies may then impact access to resources as detailed in the section on 'Resource competition'.

## Resource competition

Providing sufficient resources to animals under our care is essential to minimize competition for resources which can increase agonistic interactions, and/or prevent some animals from being able to access necessary and/or positively valued resources. It is also another way of determining the importance of a particular resource to an individual. Understanding how resource provision may affect aggressive behaviors, or understanding the impact of social structures such as dominance hierarchies has formed a large body of work across domestic livestock animals (*Krahn et al., 2023*; *Lee, Arnott & Turner, 2022*). Across a few different shrimp and prawn species, experimental tests have been conducted

to assess competition for shelter resources, food, and mates and the influence of dominance on these competitive interactions (Table 1).

Lammers, Warburton & Cribb (2009b), used the same experimental aquaria and species (*M. australiense*) as tested in Lammers, Warburton & Cribb (2009a, see 'Environmental preferences' section) to assess how refuge quality (density) and prawn size affected refuging activity patterns. In a setting with one vegetated refuge side and an open graveled side, observations showed that pairs of prawns spent more time active than single prawns in the same set-up, which also meant more time spent outside the refuge. If the prawn pairs were of different sizes, then the smaller (subordinate) prawn spent more time outside the refuge than the dominant individual (Lammers, Warburton & Cribb, 2009b). The activity was also influenced by the density of the refuge where an increasing size of the dominant individual resulted in increased activity in the high density refugia with the opposite pattern observed with the vegetation at a low density (Lammers, Warburton & Cribb, 2009b).

Dominance during competition for feed, shelter, and a receptive female has been observed in males of two morphotypes of *M. rosenbergii* (Caridea) (Barki, Karplus & Goren, 1992). Observations of groups of six male prawns showed the largest, most dominant individual was able to gain priority access to both food and shelter, although not the females, which may have been an artefact of the artificial testing scenario that prevented the typical mating rituals (Barki, Karplus & Goren, 1992). In *M. nobilii*, multiple factors affected competition for a single shelter including sex, moult stage, whether the prawns had their chela intact, the laterality of the chela, carrying eggs or not, and whether they were a prior resident of the shelter (Mariappan & Balasundaram, 2003). When *L. vannamei* were stocked at varying densities, in small groups sizes (two, four or eight individuals) in a feeding test arena, there were clear effects of both stocking density and dominance hierarchies on how the individuals behaved (Bardera et al., 2021). A higher density increased feed consumption, and a low density increased other exploratory behaviors in the arena, particularly in the more dominant individuals (Bardera et al., 2021). Feed and receptive female mate competition was also tested in *R. typus* where either two females, or two pieces of food, were presented to groups of 24 males that varied in dominance status (Dennenmoser & Thiel, 2006). Behavioral observations in the test arena showed that the degree of aggressiveness in competition for either resource depended on the dominance status, where dominant males were more competitive for females than they were for food (Dennenmoser & Thiel, 2006). Mating success, as related to dominance, altered the strategies that *R. typus* males employed to be able to gain access to a receptive female (Correa et al., 2003). In *A. heterochaelis* shrimp, mating selection was influenced by both size of the males and females, and competitive access to a shelter resource, which is necessary for successful reproduction (Rahman, Dunham & Govind, 2004). Collectively, these studies show that shrimp and prawns will compete for resources that are valuable to them, and the influence of dominance hierarchies can be assessed in a behavioral testing context in the laboratory. The value of specific resources will be dependent on the natural ecology and life stage of the species being assessed.

## Cognition and learning

Cognitive testing, including animal learning, can be utilized to understand the cognitive capabilities of a particular species or individual. In a welfare context, these assessments can be utilized to, for example, determine whether an animal possesses the capabilities to learn tasks that may be asked of them (*e.g.*, virtual fencing technology in cattle: *Lee & Campbell, 2021*), how resource provision (or lack thereof) can affect cognitive development (*e.g.*, laying hen rearing complexity: *Tahamtani et al., 2015*) or how cognitive stimulation could reduce negative, and increase positive affective states (*e.g.*, operant learning in pigs: *Zebunke, Puppe & Langbein, 2013*). Cognitive tasks, such as operant learning, can also be used to inform on the importance of a resource, such as in behavioral demand or preference testing (*Patterson-Kane, Pittman & Pajor, 2008*). To the authors' knowledge, there are only a few cognitive tests that have been applied to shrimp and prawns to date (Table 1).

In a study almost 50 years ago, *Macrobrachium acanthurus* (Caridea) individuals were tested in a Y-maze, where they were trained to visually discriminate black from white maze arms through a reward of water tank access at the end of the 'correct' maze arm (*Ventura & Mattei, 1977*). A 'punishment' was also included for half of the animals by placing them temporarily into a dry container if they made an incorrect choice. The results showed these animals were able to readily learn this discrimination task, with over 90% correct choices made across the repeated testing sessions (*Ventura & Mattei, 1977*). *Duffield, Wilson & Thornton (2015)* utilized two types of cognitive foraging tests in *P. elegans* and *Palaemon serratus* (Caridea) to understand factors that may drive innovation in these species that naturally live in highly fluctuating intertidal environments. Treatments included variation in size and variation in hunger status, with animals tested individually or in groups. The 'spatial task' required the animals to navigate through a small hole in a testing tank to indirectly reach a food source located behind a transparent barrier. The 'manipulation task' required the animals to flip over a small transparent box to reach a food source underneath (*Duffield, Wilson & Thornton, 2015*). During individual testing, hunger status did not affect task completion, but size did, with approximately half of the smaller sized prawns being able to complete the spatial task, doubling the success of the larger prawns. Neither hunger nor size affected the ability to complete the 'manipulation task', with fewer than 50% of the tested prawns obtaining the food (*Duffield, Wilson & Thornton, 2015*). Both of these studies indicate that there is capacity for these animals to display learning and willingness to complete cognitive tasks. Furthermore, they indicate that there is capability to complete tasks that may be outside ecologically relevant repertoires, as transparent barriers are unlikely to be encountered in their natural environments. Given the limited scientific evidence on cognition and learning for these taxa, there is great scope to increase application of these types of tasks to understand species' capabilities.

## Affective state, temperament, and personality

Assessing the affective state of an individual is key to understanding how they are perceiving their housing environment. This, in turn, has critical implications for measuring welfare state (*Mellor & Beausoleil, 2015*). There are also key considerations for

how responses to specific stimuli may depend on inherent differences in the animal, such as their temperament or personality traits (*Carter et al., 2013*; *MacKay & Haskell, 2015*), and, for example, how they may sit along the shy-bold axis (*Toms, Echevarria & Jouandot, 2010*). All of which play a role in welfare considerations at the individual level (*Richter & Hintze, 2019*). For decapod shrimp and prawns, there is still debate on their sentience capabilities, and thus their ability to experience emotional states such as fear, distress, pain and pleasure (*Birch et al., 2021*; *Diggles et al., 2024*). There are, however, several studies that have conducted behavioral tests to measure fear and anxiety, escape responses, and indicators of shyness/boldness using predominantly similar types of tests to those that have been validated for livestock species (*Forkman et al., 2007*). Some of these studies have also assessed correlations between different behavioral tests to measure indicators of personality within these decapod taxa. *Gherardi, Aquiloni & Tricarico (2012)* previously reviewed the literature on evidence of personality in crustaceans with only seven studies found at the time, focused on crayfish, crabs, and hermit crabs. The studies presented here detail how personality evidence has now expanded into shrimp and prawn decapods (Table 1).

*Takahashi (2022)* sought to determine if experiences affected how two species of shrimp (*Neocaridina denticulata* ssp. and *Palaemon pacificus*: Caridea) reacted in three different behavioral tests. The shrimp were exposed to a net-chasing treatment for 8 days to simulate a predator threat, relative to controls that experienced no chasing. Shrimp were then individually placed into a new tank and assessed for freezing and activity in an open field test. Following this test, vegetative shelter was added into the tank to measure latency to enter the shelter and hide as a 'shelter-seeking' test. Finally, the shelter was removed, and the shrimp were gently touched to look as 'escape-response' behavior. Results showed that net-chasing increased behaviors consistent with anxiety in *N. denticulata* only, although *P. pacificus*, regardless of treatment, exhibited freezing behavior consistent with high anxiety in the open environment. There was no effect of treatment on shelter-seeking behavior in either species, but *P. pacificus* did show greater escape responses if they had been exposed to net chasing (*Takahashi, 2022*). These results indicate behavioral plasticity in the two species where the variation in treatment effect could be related to species phylogeny and natural habitat differences (*Takahashi, 2022*). Two other studies also employed escape response testing, where *Takeuchi, Shoko & Michio (2008)* assessed for escape direction laterality in both *N. denticulata* and *Limnocaridina latipes* (Caridea) shrimp following a vibratory stimulus and *Arnott, Neil & Ansell (1999)* looked at escape responses from a predatory fish or artificial threat in *C. crangon* shrimp.

The open field test, or variants of it, have also been applied across other shrimp studies. *Harayashiki et al. (2016)* looked at post-larvae of *P. monodon* in response to dietary exposure of varying concentrations of toxic inorganic mercury. The post-larvae were tested in groups of five and given 5 min to acclimate to the open field testing environment (a gridded beaker), before being assessed for activity in the arena. While activity decreased with increases in mercury, there was no difference in time spent in the center of the open field arena which was proposed to be representative of 'risky' behavior (*Harayashiki et al., 2016*). *Rickward, Santostefano & Wilson (2024)* assessed *Neocaridina heteropoda* (Caridea)

for movement in a test tank following a 2-min acclimation while contained in a tube. After the tube was lifted, the movement behavior of the animal around the arena was observed for 4 min with predictions that bolder individuals would move more, as well as show reduced thigmotaxis (*Rickward, Santostefano & Wilson, 2024*). These tests were conducted across three repeated occasions per individual test animal. Finally, *Maskrey et al. (2018)* assessed *P. elegans* in a modified open field test as a measure of 'boldness'. Individuals were assessed three times over consecutive days in an arena that also contained a shelter. Shrimp were placed into the test arena in a plastic tube and permitted 30 s of acclimation in the tube, then 30 s acclimation out of the tube, before assessing their movement behavior in the arena for 4.5 min. Permitting acclimation to the open field test arena deviates from how this test is typically applied in laying hens, where movement responses directly following placement are quantified (*Campbell et al., 2019*; *Campbell, Dickson & Lee, 2019*). However, the open field testing can often result in freezing for the duration of the test, minimizing the ability to assess for treatment effects based on movement scores (*e.g.*, *Campbell et al., 2019*; *Campbell, Dickson & Lee, 2019*); short-term acclimation to the arena could overcome the ceiling effect issue.

Multiple different tests applied on the same individuals can allow for correlations to be assessed between tests, correlations in behaviors across time, and evidence of personality traits and behavioral syndromes (*MacKay & Haskell, 2015*). Following the open field test conducted by *Maskrey et al. (2018)*, all individuals were then tested in groups of six to determine whether their open field test behavior correlated with performance in a competitive feeding trial. A similar test arena was used, with food at one end, a shelter at the other, and observations were made of feeding time by each individual within the group. There was evidence of among-individual variation, or personality in the species, and associations between the two trials. Individuals that had been classified as more 'risk averse' and less exploratory, fed more during the competitive feeding (*Maskrey et al., 2018*). Following the open field tests applied in *Rickward, Santostefano & Wilson (2024)* 'food-shelter trials' were also applied to each individual on three separate occasions. For these trials, the shrimp were again introduced into the test arena in the cylindrical tube. After 2 min of acclimation, they were released and observed for their location at the food on one side of the arena, compared to the shelter on the other side, where bolder individuals were predicted to be out in the open for longer. The results showed significant differences among individuals in the behaviors they exhibited across the two tests. However, in general, these differences were not related to the sex nor the size of the tested individual. Inter-individual differences were consistent, but did not align with the *a-priori* predictions of fitting along the shy-bold axis. This suggests the tests may have been assessing different behaviors than what was predicted as 'risky'/'bold' for the species (*Rickward, Santostefano & Wilson, 2024*).

Finally, *Chapman, Hegg & Ljungberg (2013)* first assessed individual activity levels in a test container and then conducted a series of behavioral tests in individual *P. elegans* (Caridea) including habitat choice, exploration between two habitat choices, social shoaling preferences (as mentioned in '*Preference/choice testing, discrimination and aversion*'), and a 'fright test' to quantify boldness. The 'fright test' involved an individual

prawn locating feed in a test arena, then once they started eating, a metal weight was dropped into the tank to startle the prawn. The time to return to feeding was measured with prawns returning quicker, interpreted as being bolder (*Chapman, Hegg & Ljungberg, 2013*). This test is similar to attention bias tests that have been validated for various livestock species as a measure of anxiety (*Lee et al., 2016*; *Campbell et al., 2019*). However, for livestock testing, the stimulus is typically something that is present, then absent. This protocol makes the 'threat' unknown, inducing what would be classified as anxiety rather than fear. These attention bias tests will often look at return to feeding as a measure of anxiety. The intention to 'startle' the prawn, is also similar to 'startle tests' being validated for livestock as measure of anxiety. Although in these tests, the stimulus is again present, then absent, such as a light flash, or puff of air with the measures being the magnitude of response and time to return to a physiological baseline (*Ross et al., 2019*; *Salvin et al., 2020*). All behavioral assays were completed twice for each individual to look at consistency across time, as well as associations between all the behavioral assay measures. At the population level, activity, exploratory behavior, and fright response were consistent across time, but shoaling and habitat preferences were not (*Chapman, Hegg & Ljungberg, 2013*). At the population level, consistent behavioral syndromes were also identified with significant associations between different measures taken and differences between males and females. These sex differences align with *Bardera et al. (2020)*, who showed variation in individual behavioral profiles during a feeding test for male and female shrimp. The shrimp behaviors showed varying repeatability across days, dependent on sex and the specific behavior being exhibited (*Bardera et al., 2020*). A key evolution of welfare assessment and understanding in livestock species has been the shift from a focus on the group, to being able to track and measure behaviors and welfare impacts at the level of the individual. This has identified several important considerations around how the individual animal may vary in their use of resources within a housing system, social interactions, their temperaments, and personalities (*Richter & Hintze, 2019*). The behavioral tests to date conducted on prawns and shrimp provide evidence of consistent individual variation, equating to differing personality types which, as for livestock, could have implications for meeting the needs of all individuals in a commercial system.

## CONCLUSIONS

Behavioral testing has been applied to livestock animals for decades, providing insight into their wants and needs to guide housing structure design and husbandry practices that improve their welfare. Aquaculture of shrimp and prawns is increasing, along with recognition that the welfare of these animals needs to be considered. While it is still unclear as to the sentience of these decapod taxa, the behavioral testing applied to date across varying species demonstrates they exhibit preferences across multiple contexts, individual differences indicative of personality, cognitive capabilities, and behavioral indicators consistent with negative affective states. There is scope to increase the research focused on species of aquaculture interest, learning from what has been developed for livestock welfare assessment. Behavioral testing paradigms can be utilized to assess what physical and social resources are of importance to shrimp and prawns, including the relative value they may

place on them. Understanding of individual differences in personality traits and (pharmacological) validation of novel tests that may quantify affective state will provide tools for measuring perceptions of housing environments and impacts of management practices. The information gained from behavioral testing can guide system optimization for aquaculture shrimp and prawns in relation to the welfare of the animals.

## ACKNOWLEDGEMENTS

Thank you to G. Coman, N. Wade, and A. Rombenso for their support in learning about the shrimp and prawn aquaculture industries.

### Funding

The authors received no funding for this work.

### Competing Interests

The authors declare that they have no competing interests.

### Author Contributions

- Dana L. M. Campbell conceived and designed the experiments, performed the experiments, analyzed the data, prepared figures and/or tables, authored or reviewed drafts of the article, and approved the final draft.
- Caroline Lee conceived and designed the experiments, authored or reviewed drafts of the article, and approved the final draft.

### Data Availability

This is a literature review.

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
