# Peer review of "A review of behavioral testing in decapod shrimp (Caridea) and prawns (Dendrobranchiata) with applications for welfare assessment in aquaculture"

_PeerJ, doi:10.7717/peerj.18883_

## Round 0.1 · original submission · Major Revisions

Dear Dr. Campbell

Expert reviewers have pointed out several critical errors, notably between the title and the scope of the review. For example, keywords used in the search engine are very general, according to reviewers; they declared that you missed loads of important literature, like, for example, on thermal preferences, etc. In addition, there is no clear structure for the search of the terms or a clear method to narrow down to the main topic of the review. You should add a title to something more specific (shrimp and prawns are in some cases confused or used as synonyms), like the scientific infraorder name for them (Caridea and Dendrobranchiata). The language should be simplified to ensure that an international reader can clearly understand the manuscript. The sentences in the article are written too long and complex (less is more :) ). Therefore, there are difficulties in understanding.

You can find the comments and suggestions of the expert reviewers in the attached reports. As you will see, some reviewers recommended rejection of the MS, but I think the manuscript has a chance for revision. Consequently, a major revision is needed for your article.

I request you check and correct the manuscript according to the reviewers' reports.

Sincerely yours

Reviewer 1 ·

Basic reporting

The MS “A review of behavioral testing in decapod shrimp and prawns with applications for welfare assessment in aquaculture” is a valuable review prepared in the field. This MS can be published after corrections.
The language should be simplified to ensure that an international reader can clearly understand the manuscript. The sentences in the article are written too long and complex. Therefore, there are difficulties in understanding.
Lİne 29-33 please check the meaning and rewrite these sentences.
Line 40 “Pedrazzani et.al.” is given as Padrazzani please write the correct name.
Lİne 38-40 please check the meaning and rewrite these sentences.
Line 206 The authors have used the subtitle “Resource prevention”, instead the subtitle “Preventing access to resources” may be more appropriate.
Lİne 392-393 please check the meaning and rewrite these sentence.
Line 517 please check the word “utilis=zed”
The references list must be checked. Example in the text Line 82, the citation “Monk et al., 2021” didn’t given in references list.

Experimental design

Article content is within the Aims and Scope of the journal and article type. The study design is acceptable

Validity of the findings

The Ms includes useful information for researchers working in this field. With the necessary corrections, it will be beneficial at the international scale.

·

Basic reporting

The article is written in clear and understandable language throughout.
For the review article, an extensive literature review was conducted, and the information was appropriately evaluated within the planned subheadings.
Citations from relevant literature were appropriately referenced.
The structure of the article has been prepared in accordance with the journal's recommended format for authors.
In the framework planned for the review article, it is understood that a comprehensive literature review was conducted.
The information obtained in the evaluation of the literature is summarized in subheadings in accordance with the title (scope) of the article. The article contains collective information that can be evaluated in aquaculture activities. It also reveals the lack or insufficiency of research on which subjects and sheds light on the planning of new research. The article, which is prepared in accordance with the scope of the journal, is at a level that can attract the attention of both aquaculturists and researchers.
Although there are reviews on the evaluation of evidence of sentience in decapod crustaceans, this review examines different points and details, and highlights animal welfare.
The introduction adequately introduces the topic and clearly states the purpose of the article.

Experimental design

The content of the article is consistent with the aims and scope of the journal.
Citations have been made for the information given in the article and a reference list has been created.
There is no separate situation that requires specific ethical rules.
The method was prepared appropriately within the scope of literature evaluation.
Sources are cited and quoted (cited) appropriately.
The content of the review is organized into subsections in a coherent, sequential framework.

Validity of the findings

Conclusions are expressed appropriately.
The evaluations have been made in relation to the goals and purpose stated in the introduction. It has been explained which problems the review can answer.

Additional comments

In some paragraphs in the text rather than long quotations from the same source, I suggest supporting the topic of the paragraph with different sources. Otherwise, it is like summarizing the results of 1 or 2 research (in Lines148-163; 320-329; 373-389; 410-423).

Reviewer 3 ·

Basic reporting

The title and scope of the review is quite narrow and could be expanded to be more useful for the area of research.
If the authors still want to keep this narrow field then I would change the title to something more specific (shrimp and prawns are in some cases confused or used as synonyms) like the scientific infraorder name for them (Caridea and Dendrobranchiata) and maybe add the corresponding popular name in brackets.
This area has been reviewed recently by several authors regarding behavioural welfare indicators but mainly focused on feeding responses.
The introduction is weak and does not focus the subject clearly or explain the audience who would be directed to.

Experimental design

The search method it’s flawed from the beginning. There is no clear structure on the search of the terms or a clear method to narrow down to the main topic of the review. Because of this it does not capture the totality of the work on behaviour done on the area. Also the fact the authors only focused on shrimps and prawns and not in the whole crustaceans field, that would be more interesting and useful that just focusing on those specific two suborders, makes it quite weak in terms of usefulness. Loads have been done on crabs for example that could be added to this field of knowledge.

Validity of the findings

Because of the way the authors used the search engine the findings are very general, they missed loads of important literature like for example on thermal preferences, etc

Additional comments

The review needs a different approach, use a better survey methodology with a clear structure and also revise the different areas of study to capture all the research done.

---

## Round 0.2 · accepted · Accept

Dear Dr. Campbell

I thank you for making the corrections and changes requested by the reviewers. I read and checked your valuable article carefully and am happy to inform you that the article has been accepted for publication in PeerJ.

Reviewer 1 ·

Basic reporting

The MS “A review of behavioral testing in decapod shrimp and prawns with applications for welfare assessment in aquaculture” is a valuable review prepared in the field. The authors have made the suggested changes. The publication of the article is appropriate.

Experimental design

Article content is within the Aims and Scope of the journal and article type. The study design is acceptable.

Validity of the findings

The Ms includes useful information for researchers working in this field.